# How to Protect Free Flowing Rivers: The Bita River Ramsar Site as an Example of Science and Management Tools Working Together

Cesar Freddy Suárez [1,*], Monica Paez-Vasquez [2], Fernando Trujillo [2,*], Jose Saulo Usma [1], Michele Thieme [3], Andrea M. Bassi [4,5], Luis German Naranjo [1], Simon Costanzo [6], Oscar Manrique [7], Georg Pallaske [5] and Javier Flechas [8]

[1] Conservation and Governance Department, WWF Colombia, Cali 760033, Colombia; jsusma@wwf.org.co (J.S.U.); lgnaranjo@wwf.org.co (L.G.N.)
[2] Omacha Foundation, Bogotá 111211, Colombia; monicapaezv1@gmail.com
[3] WWF US, Washington, DC 20037, USA; michele.thieme@wwfus.org
[4] School of Public Leadership, Stellenbosch University, Stellenbosch 7602, South Africa; andrea.bassi@ke-srl.com
[5] KnowlEdge Srl, 21057 Olgiate Olona, Italy; georg.pallaske@ke-srl.com
[6] Integration and Application Network—University of Maryland Center for Environmental Science, Cambridge, MD 21613, USA; scostanzo@umces.edu
[7] Forest, Biodiversity and Ecosystem Services Direction, Ministry of Environment and Sustainable Development, Bogotá 110311, Colombia; omanrique@minambiente.gov.co
[8] Environment Planning Sub-Direction, Corporinoquia, Yopal 850001, Colombia; javierflechas@corporinoquia.gov.co
* Correspondence: cfsuarez@wwf.org.co (C.F.S.); fernando@omacha.org (F.T.); Tel.: +57-2-5582577 (C.F.S.); +57-1-2564682 (F.T.)

**Abstract:** The Orinoco river basin is the third largest river in the world by volume. Its catchment encompasses 27 major sub-basins including the Bita with a catchment area of about 825,000 ha, which originates in the Colombian high plains in the Llanos ecoregion. It has been recognized as a priority area for conservation through different gap analyses and overall determined to have good health according to the Orinoco report card 2016. The natural climate and hydrologic processes, and their synergies with flooded forests, savannas, wetlands, species diversity and local economic activities, are part of a dynamic and sensitive system. With the purpose of conserving the ecological, social and cultural benefits that it brings, the Colombian Government, with the support of regional and local civil society organizations, promoted the designation of a conservation area. Technical exercises were carried out including biological and socioeconomic surveys, local stakeholder consultations and future scenario modeling. In June 2018, the Bita River basin was designated as the largest Ramsar site in Colombia, providing a worldwide example of explicit protection of riverine systems. In order to maintain this free-flowing river, land use and fisheries management, in conjunction with other conservation actions, are being implemented and provide a model of protection for freshwater ecosystems that could be replicated elsewhere.

**Keywords:** free-flowing; freshwater; Ramsar; conservation

## 1. Introduction

The Bita River is one of the 27 hydrographic basins that compose the Orinoco River in Northern South America. The Orinoco River, with 2140 km of length, is recognized as one of the major river basins in the world. It is the third largest river basin by volume of discharge with 36,000 $m^3$/s of water flowing into the Atlantic [1]. In Colombia 11,600,000 ha of aquatic ecosystems were identified in the Orinoco in the National Wetlands Plan, in which the hydrological wealth of the Orinoco is recognized.

The Bita River basin covers about 825,000 ha, including 5070 river stretches of different hydrological orders [2]. The mainstem of the Bita is 510 km long and its waters are oligotrophic, with a slightly basic pH, highly oxygenated, with low electrical conductivity and low total solids typical of many Neotropical rivers. Along the mainstem and tributaries of this basin, fifteen types of wetlands have been identified that are fed by the temporary pulses of annual flooding [3] (Figure 1). Of particular note are the flooded forests [4] and gallery forests [5], whose soils are characterized by severe waterlogging from the blackwater rivers, alternating with periods of drought. These habitats have affinities with the seasonal várzea forests in the Amazon [4]. This basin retains 95% of its natural cover and is characterized by extensive gallery forest ecosystems along the main channel and tributaries, alternated with mosaics of non-flooded savannas and permanent and temporary flooded savannas [2] and is classified in good health condition according to the Orinoco basin health report card [6].

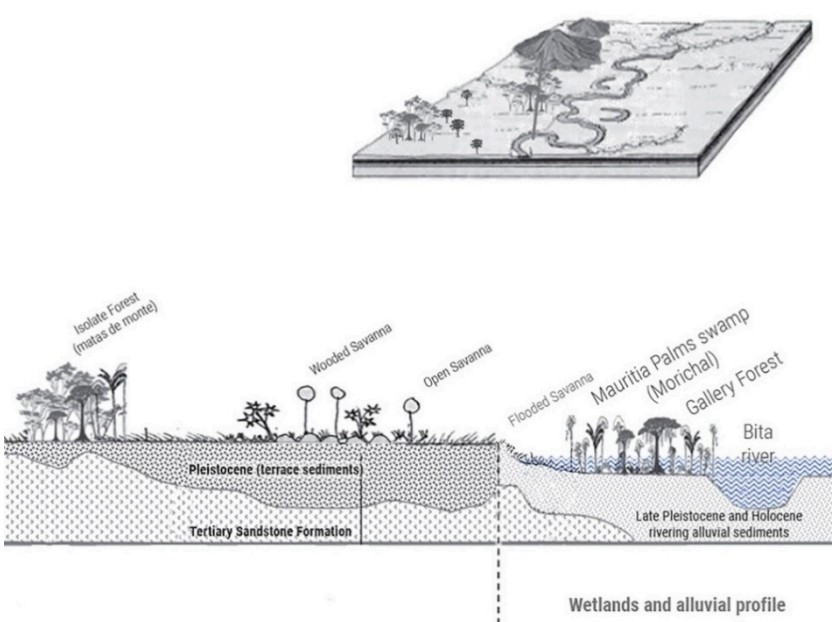

**Figure 1.** Vegetation profile representing the major riparian freshwater ecosystems [7,8].

The Bita River basin has a wealth of biodiversity represented by: 1474 species of plants [9], 87 species of aquatic macroinvertebrates [10], 34 species of dung beetles [10], 10 species of shrimp and one crab [11], 254 species of fish [12], 29 amphibians, 58 reptiles [13,14], 201 birds [15] and 63 mammals [16]. Nineteen of these species are listed as in some degree of threat of extinction [17].

Those who depend on these ecosystems for their livelihoods are mainly local communities that live along or near the river and nomadic collecting groups of indigenous people belonging to the Sikuani, Piapoco, Puinave, Piaroa, Curripaco, Saliva, Cubeo, Cuiba and Amorúa ethnic groups, who carry out activities associated with hunting and fishing.

This biological richness and the associated ecosystem services have facilitated the identification of this region as a high priority for conservation [18–20]. In this paper we present the process that has been undertaken to achieve its designation as a Wetland of International Importance [21] by the Colombian Government and the Ramsar Convention on Wetlands in 2018. At the same time, we mention the current and further challenges to implement proper conservation actions through different management instruments. This is the first time that an entire river basin has been designated as a Ramsar site within South America. It is an opportunity to assure the persistence of its conservation values and ecosystem services and maintenance of its status as a free-flowing river.

## 2. Methods

The work to establish environmental sustainability of the Bita River began in 1995 and continues to this day (Figure 2). Initially, a baseline of knowledge was established including studies of biodiversity, wetlands, fisheries and ecotourism supported by the Tropical Forest Conservation Act—TFCA [22–25]. This information identified this region as an important area for biodiversity conservation under various criteria and methods. For instance, using an expert opinion [19], the first binational priority conservation areas exercise highlighted the Bita River as one of the priorities for biodiversity protection in the entire Orinoco basin. This was subsequently corroborated using systematic conservation planning tools [20,26]. In 2014, the Alliance for the Protection of the Bita River was formed with government entities such as Corporinoquia, National Parks of Colombia, the Vichada Government, the National Navy of Colombia and the Alexander von Humboldt Institute with the mission of design and develop a rural strategy to protect the Bita River [27]. At the same time, civil society organizations also joined the Alliance including the Omacha Foundation, the Orinoquía Foundation, La Pedregoza Environmental Corporation, the Palmarito Foundation and WWF Colombia.

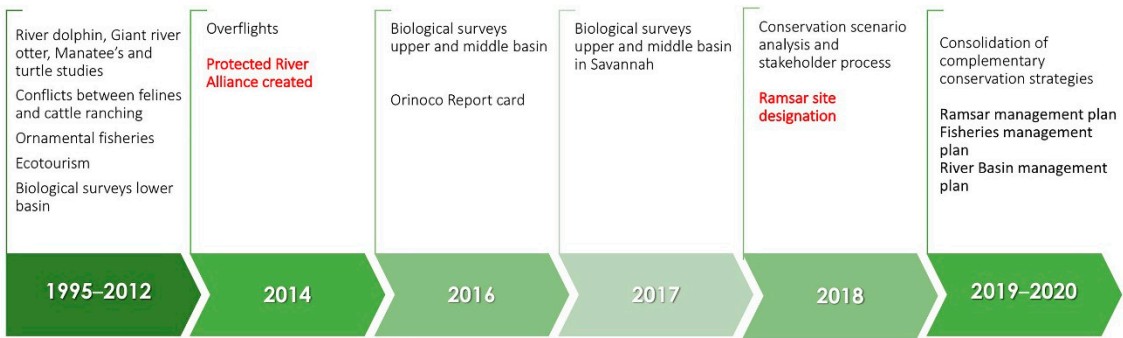

**Figure 2.** Conservation timeline.

Between 2015 and 2017 several biological studies of the Bita River [17] and an analysis of the socioeconomic aspects related to basin health [6] were carried out. A basin health report card was also completed for the entire Orinoco basin in 2016. The report card is a tool that synthesizes indicators across social, environmental and economic domains related to basin health and is used to describe ecosystem status, increase public awareness, and inform and influence decision-makers. For the Orinoco and its sub-basins, including the Bita, eleven indicators were evaluated by comparing data for each relative to scientifically derived thresholds (Table 1). This resulted in scores for each indicator in each river basin. Scores for each indicator were averaged by category of indicators, and then averaged across to arrive at one overall score for each sub-basin. Report card grades were assigned based on this average score for each sub-basin [28].

In the Orinoco report card the Bita basin had a grade of B, equivalent to an overall good health score, meaning that the social, environmental and economic indicators for the basin were presently, on average, in good condition. However, it was necessary to develop a better understanding of how the system components would change under a range of alternative development options for the basin.

We made use of a system thinking (ST) approach that at the same time used causal loop diagrams (CLD) to analyze and identify the main driving forces to determine the relationships between different agents (ranchers, farmers, fishermen, forest plantations owners, tourism organizations and local governments) and subsystems (land use, economy, water, biodiversity, tourism, fisheries, population and governance). Governance context analysis and participatory system dynamics modeling were used to visualize the interdependencies between the complex social-ecological systems in the Bita River basin. In order to do that, a system thinking (ST) [29] approach was applied to better understand the complexity of the system. Practically, this means analysis was completed to understand and quantify the

extent to which the social, economic and environmental dimensions of development interact at the local level. Examples include how population growth affects natural resource use, the extent to which the extraction of resources results in short term income creation, and, consequently, how the declining quality of the environment is impacted by socioeconomic activity, trade-offs that lead to reduced income creation in the medium and longer term.

**Table 1.** Orinoco River report card indicators, values and scoring system [28].

| Indicator | Value | Scoring System | Grading System |
|---|---|---|---|
| Water quality<br>Risk to water quality<br>Water supply and demand | Water | | |
| Natural land cover<br>Stable forest area<br>Fire frequency<br>Terrestrial connectivity<br>Ecosystem services | Ecosystems and Landscapes | 0–100% | F-A |
| Human nutrition | People and Culture | | |
| Mining in sensitive ecosystems | Management and Governance | | |
| Dolphin abundance | Biodiversity | | |

ST was chosen because it is a methodology that supports the identification of key variables in a system (e.g., water quantity, water quality, number of sport-fishing tourists, or the estimation of the fish stock). It does so with a group model-building approach, where 30–40 participants can cocreate a system map, also called causal loop diagram (CLD) (see Figure 3). The use of ST and group CLD modeling techniques support a deeper understanding of behavior, resulting impacts and emerging trends. This also creates a shared understanding among stakeholders, as it results from the interaction of experts, practitioners and citizens with different knowledge, expertise and goals. CLDs also allow for the integration of inputs from stakeholder consultations with known dynamics (cause–effect) or patterns of behavior from the scientific literature [30–32]. Local communities from Puerto Carreño and La Primavera (the main municipalities with influence in the basin), local authorities and agricultural, forest plantation and ecotourism organizations and academics, government representatives and civil society organizations participated in the process during five workshops held in 2018.

The use of ST, and the creation of CLDs served as the blueprint for the creation and customization of a system dynamics (SD) model. In fact, the model presented in this paper was fully customized to the local context, being informed by the group model building exercise. No "off the shelf" model was employed, given the unique nature of the Bita River basin. The modeling process was carried out using the software Vensim. The model was parameterized using quantitative and qualitative data sources from public and non-public data, workshops, surveys and interviews. The model has been presented in detail in other publications [33] and uses stocks and flows (e.g., to represent natural resource and built infrastructure stocks and their dynamics), feedback loops (to understand how changes in one part of the system affect others), delays (to better understand how changes in socioeconomic performance influence natural systems, which normally respond with different, and often longer, timeframes) and non-linearity (to better capture the extent to which different parts of the system are interconnected).

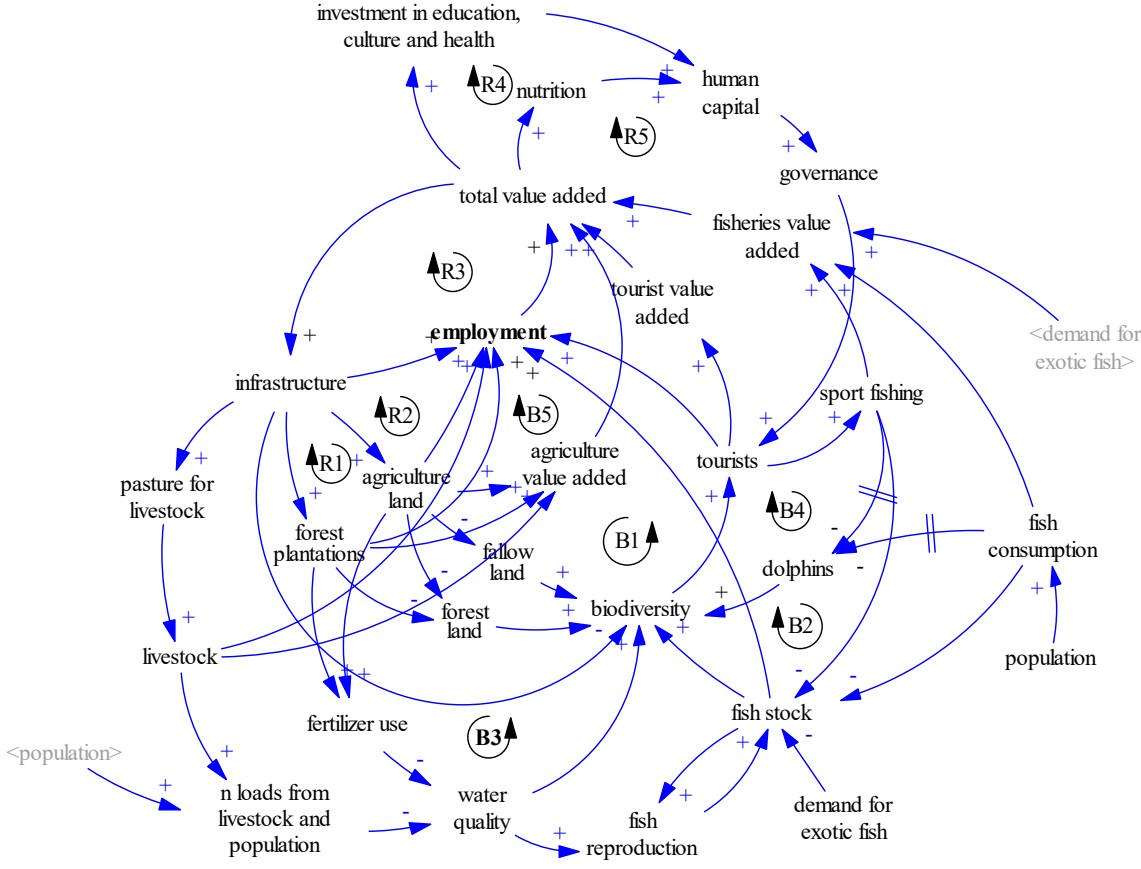

**Figure 3.** Causal loop diagram or system diagram of the Bita River basin.

Further, the shared understanding created during the group model building sessions supported the formulations of scenarios. This is a process in which the expectations and desires of several local stakeholders were formalized in storylines and then in numerical model input for the creation of forward-looking simulations. The key indicators to use in the model to assess performance under different scenarios and to analyze and validate simulation results were identified during the aforementioned stakeholder workshops in 2018. All stakeholders are inhabitants of the Río Bita, and voluntary conservation agreements were also discussed.

These are the five main scenarios that were developed and analyzed for the Bita basin [23]:

1.  Pessimistic: assumes a growth of existing patterns of resource use, with no changes to the economic structure of the area. In addition, it assumes large population growth, driven by immigration from Venezuela.
2.  Business As usual (BAU): assumes that the Bita basin will continue to follow historical trends (e.g., for population and land cover changes). This is primarily due to the underlying assumptions that policies and development strategies will not change.
3.  Ramsar site: assumes the implementation of a Ramsar site, that is an international designation under the Convention on Wetlands of International Importance where the contracting parties (countries, in this case the Colombian Government) are expected to manage and to maintain their ecological character and retain their essential functions and values for future generations and hence focuses on reducing the exploitation of natural resources and on curbing the historical trend of land cover change. Nevertheless, the establishment of the Ramsar site does not imply full conservation [34]. In fact, economic activities are allowed, but these should not cause harm to the environment. In this scenario we are assuming a proper management of natural resources without harm to it.

4. Balance: assumes a balanced development between BAU and strict conservation. This scenario has more land use restrictions than the Ramsar site case, maintaining low human population levels (to support economic development) and allowing limited land cover change (to emphasize sustainability). This scenario implies the creation of a protected area in IUCN category IV.

5. Conservation: assumes efforts to protect the Bita River basin, through stricter rules and regulations that aim at improving natural capital, and implies the creation of a protected area in IUCN category I or II (these actions include land use restrictions, prohibition of impact activities as fauna and flora hunting, mining and oil extraction, infrastructure construction and land ownership limitation to reduce human impact and promote direct terrestrial and freshwater ecosystems preservation).

Finally, a process was begun to develop and coordinate implementation of three management planning instruments for the Bita River Basin—the Fisheries Management Plan, the River Basin Management Plan (POMCA after its Spanish acronym) and the Ramsar Site Management Plan.

## 3. Results and Discussion

### 3.1. The Bita River as a System

In the Bita River basin the relationships between nature and humans are complex and the many variables that can be used to assess social, economic and environmental performance are heavily interconnected (see Figure 3). For example, economic growth is based on the recent expansion of forest plantations, agricultural land, fishing and ecotourism, which all generate employment. However, inadequate management practices affect the amount of natural ecosystems and alter water quality, both of which underpin economic activities in the basin. Fires are usually related to poor land management practices, and climate change is expected to increase their frequency and severity. The extent of forest lands and savannas and the state of water quality are important metrics of ecosystem integrity and their status affects the state of biodiversity. Tourism and fisheries during the dry season are also strongly affected by freshwater biodiversity status, including the quality of the environment. Tourism, primarily focused on sport fishing, generates income, which also comes from employment in agriculture, fisheries (other than sport fishing) and determines the amount of funding available to invest in culture, education and improved nutrition. Culture, education and nutrition also affect governance, which impacts on human action over natural resources (e.g., on fishing practices and on the degree to which people influence the environment). These interconnected dynamics form circular relations, or feedback loops. These feedback loops have shaped the historical pattern of development of the Bita River and will continue to do so in the future.

Key Feedback Loops in the Bita Model

The CLD for the Bita River basin represents the complexity of the relationships among social, economic and environmental variables in the system and presents these relations or feedback loops as arrows between variables (Figure 3):

- There are two types of relationships, or causal relations, between variables: positive or direct (+), if both variables (cause and effect) have the same behavior (increase or decrease); and negative or inverse (−), when the cause–effect variables change in opposite directions. If the cause variable increases (decreases), the effect variable decreases (increases).

- Interactions between variables can generate feedback loops. These can be reinforcing (R) or balancing (B) loops. Balancing feedback loops counter change, and represent constraints, or tradeoffs, in a system. Reinforcing feedback loops represent the notion of unconstrained exponential growth or decline. The interaction of balancing and reinforcing feedback loops, and the respective strength of their effects, determines the behavior of the system.

The reinforcing loops, which cause growth, or the expansion of economic activity, employment and income creation in the CLD, include:

- R1, R2 and R3, showing that economic growth leads to more resource use (e.g., land), which, by generating income and profits, further stimulates economic growth;
- In addition, the growth of GDP leads to social and cultural empowerment (R4 and R5), increasing human capital and stimulating economic growth further;

Practically, the model contains five reinforcing loops (R1 through 5) that capture the beneficial effects of increasing economic activities and income. These benefits can be seen in effects of the expansion of infrastructure and economically productive land (R1 and R2) on employment (R3), education, culture and health (R4) and nutrition (R5). Subsequently, human capital is expected to improve, with the potential growth of tourism and investments also improving. Specifically, an increase in value added and wealth leads to higher amounts of money available for investments and nutrition, which increases human capital in the long run. Human capital improves governance and hence the reputation and safety of the area and the use (and conservation) of natural resources, which makes it more attractive for tourists to visit and more sustainable for local economic activities. A virtuous cycle is initiated, which draws more tourists into the area, thereby increasing the total value added, which ultimately increases (a) disposable per capita income for nutrition and (b) investments in education, culture and health (although these are constrained by the relative isolation of the region from the main economic areas of the country).

Further, the economic growth triggered by reinforcing loops (e.g., through farming and forest plantations) comes at the expense of natural capital, triggering balancing loops:

- B1 through B5 in the CLD show two impacts: (1) depletion of natural resources and (2) reduction in the quality (or health) of ecosystems. In fact, (1) land cover changes may lead to a decline of biodiversity, possibly reducing tourism arrivals, and hence employment and income; and (2) the use of fertilizers and pesticides, negatively impacts water quality and fish stocks.

As an example of the dynamics triggered by balancing loops, loop B1 is a balancing feedback loop that affects biodiversity, tourism, infrastructure and total value added. The Bita River basin is a nearly pristine area rich in natural land cover and biodiversity, which makes it attractive for tourists to visit, especially fishermen. Tourists, in turn, generate revenues for the local government and population, which can be used for subsistence or reinvested. It is assumed that a fraction of the total income is used for the expansion of infrastructure. However, in the Bita basin, which is largely untouched and remote, the expansion of infrastructure is likely to come at the expense of natural land cover and hence negatively affect biodiversity. This implies that infrastructure development has the potential to undermine tourism growth. In fact, if the effect of infrastructure expansion on biodiversity is too strong, it might reduce the annual number of tourist arrivals.

More details on the model and all key feedback loops are presented in Bassi et al. [34].

*3.2. The System under Threat and Interactions between Drivers of Change*
3.2.1. Fires

Fire frequency and intensity is currently one of the major threats to biodiversity in the basin. During the dry season of 2020 more than 100,000 ha were burned. The Bita report card [6] uses an equation for fire frequency, anchored on historical data and accounting for the trend of natural land cover, as the number of fires increases as land conversion is more intense. A threshold of 200 fires was identified comparing multitemporal fire frequency and through stakeholder consultations (land owners, farmers and local authorities) during a workshop held in Puerto Carreño, Colombia. Participants indicated that about 20% of fires are natural and the remaining 80% are man-made, and hence are possibly affected by the improvement of land management practices.

In the Bita basin, the areas that present the highest risk of fire occurrence correspond to the undulated savannas, which is the most common land cover type and is classified as

being of high to very high-risk for fires, followed in risk level by savannas, and permanently flooded savannas. There is a higher incidence of fires within one kilometer of the rivers and that incidence increases moving closer to the river. Similarly, about 90% of the fires occur within the 6–7 kilometers from roads. The highest concentrations of burned areas also occur in the first months of the year and coincide with the times of low precipitation (January, February and March). Fire in this basin is correlated with topographic, vegetation, soil type, climatic (e.g., surface temperature and precipitation) and anthropic factors (e.g., human settlements and roads) [35].

### 3.2.2. Fishing Activities

Ornamental, sport and commercial fishing constitute the second income after cattle ranching, which puts pressure on specific species whose commercialization feeds the markets of the cities of Villavicencio and Bogotá [24]. Stingrays, arawanas and loricariids with high commercial value in international markets are harvested in the Bita River basin. Sport fishing activities are carried out with selective fishing gear. Annually, more than 1000 sport fishermen visit between the months of December to March (low water). Aquaculture production has recently begun in the basin (Arawana azul *Osteoglossum ferreirai*) and may become an alternative to traditional fishing, although it is still in an experimental phase [36]. Sport fishing attracts tourists for ecotourism activities and provides an important new source of revenue. Under the assumption that fish from sport fishing activities are returned to the river, this activity has minimal impacts. However, any increase in the current level of fish catch (ornamental and sport fishing) will likely contribute to an overexploitation of fish stocks and a reduction of species richness (or of the mean size of the fish that are caught). The depletion of fish stocks would undermine the potential for sport fishing activities and consequently reduce the attractiveness of the Bita River region for tourists.

The health of fisheries depends on the climate and natural flow regime, and threats include habitat destruction, overharvesting and pollution [37]. Between May and November, the spawning season of most fish occurs, when the river has higher flows due to the increasing rains, and the number of fishes caught for export is at its minimum. In general, farmers are also fishermen, who in times when the flow of the river decreases, use the floodplains for agricultural activities, to grow food for subsistence and on occasion for additional income.

### 3.2.3. Infrastructure

The Bita River basin is a nearly pristine area with mostly intact natural land cover and rich in biodiversity. Tourists in turn generate important revenues, which can be used for subsistence or reinvested in other economic activities as agricultural systems. It is assumed that a fraction of the total income is used for the expansion of infrastructure. However, in the Bita area, the expansion of infrastructure, such as the road between Puerto Carreño and Puerto Gaitan, is likely to come at the expense of natural land cover and hence negatively affect biodiversity. This implies that too much or poorly designed or sited infrastructure development has the potential to undermine tourism growth, despite the connectivity improvements.

### 3.2.4. Land-Use and Land-Use Change (LULUC)

Most farmers in the basin are fishermen who, use the floodplains and riparian areas as agricultural fields during the low water season. However, agriculture, forest plantations, cattle ranching and infrastructure are expanding, enabling the local population to expand land use for economic activities and hence reduce natural land cover. For instance, in the case of forest plantations, they are mainly being developed by national and international companies and, at present, account for almost 14% of land use change within the Bita basin. The main tree species grown in the basin are pine, eucalyptus and acacia. Changing land use practices to forest plantations leads to an increase in the use of chemical fertilizers, which negatively affects water quality and ecosystem health. A decrease in water quality

puts additional pressure on the fish stocks, poses threats to biodiversity and reduces the attractiveness of the area for tourism.

### 3.2.5. Climate Change

The Bita basin has been identified as one of the most vulnerable regions to climate change in the Orinoco basin, especially associated with anticipated increases in water stress due to increased droughts [38]. The Bita River basin is dominated by the ultisol soil type, which is characterized by high clay content and low permeability. During rain events, these poorly permeable soils result in high levels of run-off and associated erosion.

### 3.3. Future Scenario Modeling

The results of the scenario modeling show the potential conflicts from extensive population growth and land use change in the Bita basin with the integrity of the ecosystems in the area and sustainability of the tourism industry. A growing population, ornamental fish catch for export and sport fishing, in addition to the discharge of pollutants in the river put pressure on fish stocks and tourism revenues. Initially, the fish stock is projected to decline in all scenarios. The pessimistic scenario shows the strongest decline as a consequence of the rapidly growing population, which causes an overexploitation of the fish stock. Catch levels depend on demand and on available supply (i.e., the stock of fish). In the BAU and the pessimistic scenarios the rapid increase in annual catch exceeds the natural reproduction rate. Catch levels in both scenarios are high and unsustainable in the short term, which causes the overexploitation and the decline of the fish stock. As a consequence, total catch levels decline in both scenarios in the medium and longer term. In the conservation scenario instead, the catch level increases moderately.

The pressure on fish stocks and the potential expansion of infrastructure, generate additional reduction for natural capital and biodiversity. The model indicates that, if current trends continue (BAU scenario), the Bita area will lose about 70% of its biodiversity by 2040 due to the forecasted socioeconomic development (Figure 4). In the pessimistic scenario more than 90% of the biodiversity in the Bita basin would be at risk of being lost. Conservation practices help to maintain biodiversity and significantly contribute to improving ecosystem health and resilience, as presented in the conservation case.

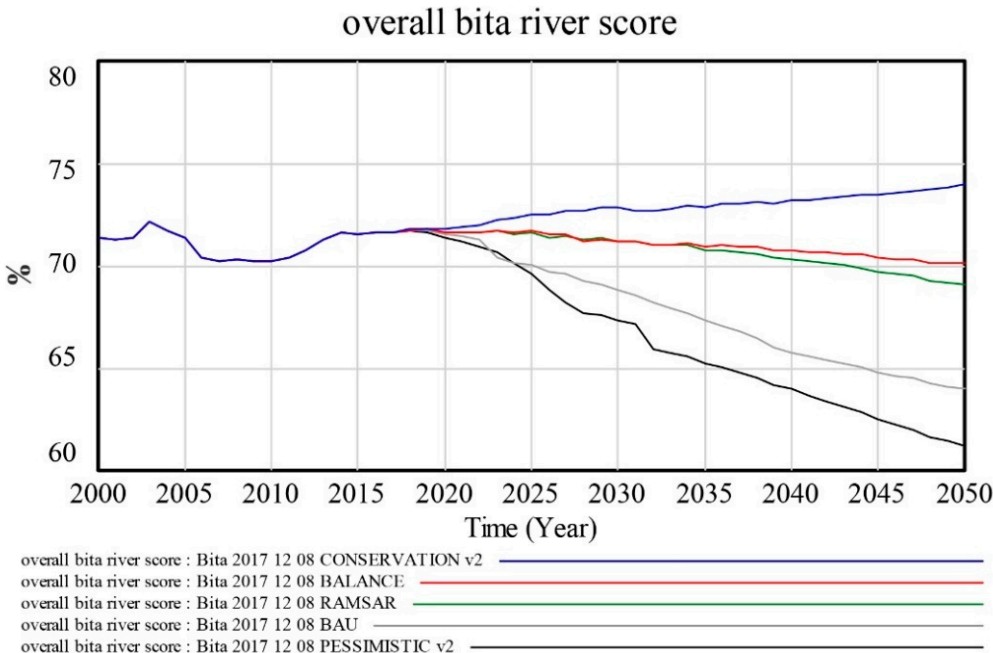

**Figure 4.** Conservation scenarios for the Bita river basin with modeled overall scores for the basin [33].

Human-induced environmental pressures and worsening water quality are likely to impact the river dolphin population in the Bita area. Dolphins are sensitive to water quality and fishing activities [39], which indicates that river dolphins in the Bita might migrate if the situation worsens. The development in the pessimistic and the BAU scenario lead instead to a rather rapid deterioration of environmental quality, as indicated by the outmigration or decline of river dolphin populations, this corroborates information reported by Mosquera-Guerra [40], which estimated a decrease of 35.7% in the number of individuals in six years for the Colombian Orinoco.

More detailed results of scenario analysis are presented in Figure 5, with color coding to indicate desirable (green) and undesirable (red) scenario outcomes.

| | Scenarios | | | | | | | | | |
|---|---|---|---|---|---|---|---|---|---|---|
| **Indicator** | **Pessimistic** | | **BAU** | | **RAMSAR Site** | | **Balance** | | **Strict Conservation** | |
| | Short term | Long term | Short term | Long term | Short term | Long term | Short term | Long term | Short term | Long term |
| **Economy** | | | | | | | | | | |
| Value added | 3.04% | 3.73% | 1.47% | 1.05% | 1.10% | 0.97% | 0.75% | 0.39% | 0.81% | 1.31% |
| Value added per capita | -0.80% | -0.33% | 0.21% | 0.04% | 0.06% | -0.03% | 0.17% | -0.11% | 0.70% | 1.31% |
| **Population & tourism** | | | | | | | | | | |
| Population | 4.1% | 4.1% | 1.0% | 1.0% | 1.0% | 1.0% | 0.5% | 0.5% | 0.0% | 0.0% |
| Tourism arrivals | -0.1% | -0.7% | 1.0% | -1.5% | 2.8% | 0.0% | 1.9% | -0.7% | 3.5% | 5.0% |
| **Land cover** | | | | | | | | | | |
| Forest plantation | 4.9% | 3.6% | 2.0% | 1.9% | 3.1% | 2.7% | 0.7% | 0.0% | 0.7% | 0.0% |
| Agriculture land | 2.1% | 3.0% | 3.1% | 2.9% | 1.0% | 1.0% | 0.5% | 0.5% | 0.0% | 0.0% |
| Pasture land | 0.9% | 0.3% | 0.0% | 0.0% | 0.0% | 0.0% | 0.0% | 0.0% | -2.1% | -2.0% |
| Settlement land | 5.1% | 3.6% | 1.0% | 0.8% | 0.9% | 0.8% | 0.6% | 0.4% | 0.3% | 0.0% |
| **Agriculture practices** | | | | | | | | | | |
| Fertilizer use | 5.1% | 5.0% | 4.9% | 5.0% | 1.0% | 1.0% | -2.6% | -2.4% | -2.4% | -2.1% |
| Fires | 6.3% | 12.3% | 3.0% | 3.0% | -0.1% | 0.0% | -0.1% | 0.0% | -4.8% | -1.7% |
| **Freshwater ecosystem** | | | | | | | | | | |
| Fish catch in the Bita | 5.4% | -1.1% | 3.9% | -1.3% | 0.2% | -0.7% | 1.5% | 0.7% | 1.1% | 2.2% |
| Bycatch | 4.0% | 2.2% | 0.9% | 0.4% | 0.1% | 0.1% | -0.5% | 0.1% | -1.1% | -0.7% |
| Dolphins | -1.1% | -4.4% | -0.7% | -2.3% | -0.3% | -1.4% | -0.2% | -0.7% | -0.2% | 0.3% |
| Infrastructure | 2.9% | 3.1% | 1.8% | 1.8% | 0.2% | 0.6% | 2.7% | 2.4% | 0.1% | 0.8% |
| Protected areas | 0.0% | 0.0% | 0.0% | 0.0% | 3.3% | 2.9% | 3.3% | 2.9% | 5.2% | 3.0% |

**Figure 5.** Overview of the annual rate of change for selected indicators. All scenarios. Legend, color coding: Green: desirable outcome; Red: undesirable outcome [33].

The results of the modeling process, the discussions that emerged in the workshops and direct conversations with local stakeholders contributed to a shared understanding of the economic, social and environmental beneficial or detrimental outcomes of the scenarios. The Ramsar Site scenario did not have a negative trend in economy and had positive changes in land use reducing pasture lands, fires and fisheries. This transparent and participatory approach resulted in the endorsement of the results, and the positive concept to design the Bita River with an international figure. However, these results show the need to carry out different approaches in the implementation to improve the sustainability of the Bita River.

*3.4. Governance and Political Process for Achieving Designation of the Bita Basin as a Ramsar Site*

The process of designating the Bita basin as a conservation area followed on recognition of the need for additional protection of savanna grasslands in the national protected area systems as identified by gap analyses. The national government under the leadership of the environmental ministry included the Bita River into the Ramsar portfolio after extensive stakeholder consultation and with the support of the Omacha Foundation and WWF. First, a series of fauna and flora studies played a key role to identify and fulfill the Ramsar

convention criteria 1 (sites containing representative, rare or unique wetland types), criteria 2 (sites that supports vulnerable, endangered or critically endangered species or threatened ecological communities), criteria 3 (sites supporting populations of plant or animal species important for maintaining the biological diversity), criteria 4 (sites that supports plant and/or animal species at a critical stage in their life cycle, or provide refuge during adverse conditions) and criteria 8 (area important source of food for fishes, spawning ground, nursery and/or migration path on which fish stocks, either within the wetland or elsewhere, depend on). Then, a participatory process was completed to evaluate the social, cultural and economic values. Local communities from Puerto Carreño and La Primavera (the main municipalities with influence in the basin) and local authorities and agricultural, forest plantation and ecotourism organizations discussed the future scenarios and potential solutions to stop degradation and promote sustainable use during five workshops. The system dynamic modeling exercise was used during the workshops to understand the relationship between different components of the Bita River system and to discuss the best conservation scenario to consider to preserve the natural values without compromising sustainable economic development. There was also significant discussion and clarification on questions regarding the Ramsar designation, land ownership and land use implications. Once all the technical information was compiled for the Ramsar application and the legal aspects evaluated by the environmental ministry, the Ramsar site was designed in June of 2018.

After the Ramsar designation, the management planning process and coordination between three management instruments began; these are the Fisheries Management Plan; the River Basin Management Plan (POMCA after its Spanish acronym) and the Ramsar Site Management Plan. All of them work together to ensure actions that promote sustainable management of the river basin. Under this framework and with the aim of ensuring the maintenance of ecological characteristics of the Bita River basin, a management plan for the Ramsar site is being formulated under the leadership of the Colombian Ministry of Environment, the regional environmental authority—Corporinoquia, and with the coordination of the Omacha Foundation and all actors involved in the process. The Regional Wetlands Committee of the Bita Ramsar site is also being created.

The Ramsar Site Management plan seeks to promote conservation, rational use and capacity building of locals for management and use of the natural resources in the area. At the same time, the POMCA is under development, which focuses on water resources management that balances economic uses and conservation in the basin. Under the direction of the National Aquaculture and Fisheries Authority (AUNAP), with the support of the Orinoquia Foundation and Omacha Foundation, a Fisheries Management Plan is also under development. It includes an assessment of current fishing activity, actors and socioeconomic conditions. The management plan will promote the sustainable extraction of ornamental fish, the practice of sport fishing (catch and release) and restrictions on the use of fishing nets in the basin. Additionally, a long-term monitoring and surveillance program is envisaged, which will engage local actors.

These three management instruments are being formulated through a series of participatory exercises with local landowners, forest companies, fisherman representatives, local governments and ecotourism agencies. The results include specific programs and projects aimed at conserving, preserving, protecting or preventing the deterioration and/or restoration of the basin. A zoning scheme has also been elaborated (Figure 6). It includes planned conservation actions on 601,112 ha, sustainable use in 92,358 ha and restoration actions over 124,544 ha.

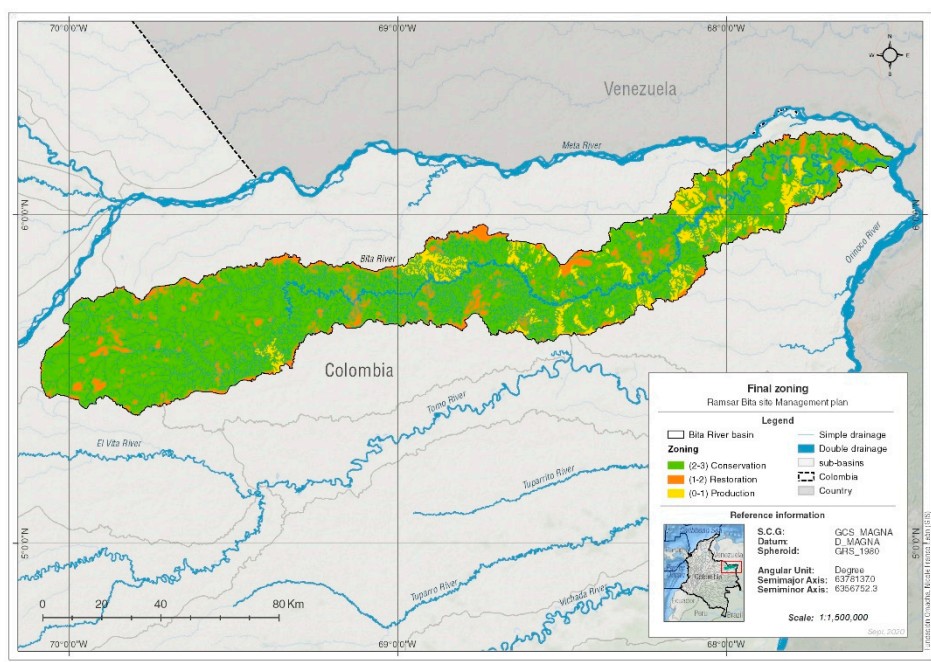

**Figure 6.** Conservation zoning of the Bita River Ramsar Management Plan.

*3.5. Main Conservation Actions and Challenges*

3.5.1. Conservation Actions with Target Species

One year after the Ramsar site designation, a biological corridor with 228,457 ha was established as a result of the ongoing planning processes. Many conservation agreements and actions have been developed. These agreements are focused on eight groups of representative animals of the basin, such as tapir (*Tapirus terrestris*), jaguar (*Panthera onca*), puma (*Puma concolor*), river dolphins (*Inia geoffrensis*), otters (*Pteronura brasiliensis, Lontra longicaudis*), turtles (*Podocnemis* sp.), stingrays (*Potamotrygon* spp.) and the peacock bass (*Cichla ocellaris*). In addition, the Wildlife Conservation Society (WCS) is supporting conservation agreements with tapirs (*Tapirus terrestris*), morichales/palm swamps (*Mauritia flexuosa*), congrio (*Ascomium nitens*) and saladillo trees (*Cairapa llanorum*). In addition, La Pedregoza Environmental Corporation has developed a seed recovery, nursery management and propagation program for the congrio and saladillo. This program has made conservation agreements with properties in the Bita basin.

The agreements also require compliance with the legal provisions issued by the environmental authorities in relation to the hunting of wild fauna, conservation of the lateral river connectivity, the closure season for harvest of certain species, regulations regarding the commercial production of forest resources, the development of responsible fishing practices such as suitable fishing gear, respect for closed periods, minimum fishing sizes and the implementation of good practices in tourism activities related to dolphin watching and sport fishing. In turn, the agreements incorporate good practices in livestock production focused on improving production to mitigate the conflict with big cats (puma and jaguar), this includes improving sanitary conditions (good management of cattle ranching), avoiding hunting for retaliation and prey (when big cats attack cows and other farm species), preserving natural conditions of the felines, the confinement of vulnerable cattle in pens near houses and the installation of drinking troughs and salting troughs away from water sources, among other measures.

3.5.2. Conservation Agreements with Foresters and Other Local Actors/Partners

As a result of the work with the local landowners in the basin over the last two years, it has been possible to formalize six conservation agreements in 147 properties that cover a total area of nearly 200,000 ha in the basin and some surrounding areas, of which

115 properties are located strictly within the basin with an area of 89,855 ha. These sites include private plots with 30,750 ha, and forest plots with 168,179 ha.

Additionally, these sites are implementing good practices for forestry production through actions such as the control and management of herbicides and agro-inputs, the control of exotic species in native forests, the implementation of environmental management plans, the prevention of and fire management and training in conservation and management issues.

### 3.5.3. Further Activities to Implement

Through the implementation of the Ramsar Management Plan, the aim is to maintain and improve the physical biotic conditions of the basin, through the development of concrete actions established in different strategic lines. In this regard, complementary conservation agreements with local owners are needed to conserve species and ecosystems. A clear example is the creation of private reserves or the implementation of conservation actions for target species (Section 3.5.1). These conservation agreements should be implemented in conjunction with participatory monitoring programs, where physical conditions such as land cover, climate or water resources are measured. At the same time, a set of activities focused on fire management and the creation of an early warning system are planned. Governance and coordination of the defined strategies, decision-making processes and actions to solve problems of common interest in the basin, particularly those related to water management, are also a critical needed piece of work. All those activities should include the promotion of sustainable production incorporating climate change adaptation, especially in respect to the main economic activities (i.e., livestock, sport fishing, ornamental and commercial fisheries).

### 4. Conclusions to Keep the Bita River as a Healthy, Free-Flowing River

To maintain the good health of the Bita River basin, building on the work that has already been done, it is necessary to implement the following actions:

- Monitoring and research: there is a need to continue ongoing monitoring and to expand it to ensure that the goals of the management plans are being met over time. Monitoring tools and frameworks should include a set of biodiversity attributes (species composition, structures and functions) and tracking impacts in the social and economic dimensions.
- Conservation agreements: effective conservation for the Bita basin does not end with the designation as a Ramsar site. The management plan and the implementation of local conservation agreements are critical for long-term conservation and sustainable economic development of the basin.
- Improved governance around land management: the participation of all stakeholders at different levels in designing, creating and implementing conservation allows increased knowledge of the decisions that may have an impact on natural resources and ensures that all involved have a voice in shaping management actions. Participation by diverse stakeholders in those decisions is a critical part of the sustainability of the management plan implementation, including those related with financial aspects. Coordination between national, regional, local governments and civil society organizations should generate conditions to implement sustainable initiatives and improve the effective management of the designated conservation area.
- Implementation of sustainable production systems and good management practices: fisheries and agricultural–livestock systems are a source of employment and incomes in the basin. However, the performance of these economic activities can also have negative impacts on natural systems and that, in turn, can cause reduction on production (one of the key reinforcing loops identifies). For this reason, it is critical to expand and strengthen the implementation of best practices in fisheries and agricultural–livestock management.

- Capacity building: Academic, research institutes and technical teams should interact with local communities and regional educational platforms to spread the knowledge about sustainable uses and best practices of production. Monitoring and research initiatives can include young people and local communities and provide a higher likelihood for a future with a healthy, free-flowing Bita River.

**Author Contributions:** Conceptualization F.T., M.T., J.S.U. and C.F.S.; methodology F.T., A.M.B. and S.C.; formal analysis G.P., M.P.-V., C.F.S.; writing—original draft preparation C.F.S. and M.P.-V.; writing—review and editing C.F.S., M.T., F.T., L.G.N., A.M.B. and J.S.U.; visualization C.F.S. and M.P.-V.; validation J.F. and O.M. All authors have read and agreed to the published version of the manuscript.

**Funding:** This paper brings together various efforts to ensure freshwater ecosystem conservation along the Bita river funded by the Tropical Forest Conservation Act in Colombia (TFCA), WWF US, WWF NL, Corporinoquia and participation of the Luc Hoffmann Institute through the LIVES project on the modeling process.

**Institutional Review Board Statement:** Not applicable, because this article does not contain any studies with human or animal subjects.

**Informed Consent Statement:** Not applicable.

**Data Availability Statement:** Not applicable.

**Conflicts of Interest:** The authors declare no conflict of interest.

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
