# Peer review of "How to Protect Free Flowing Rivers: The Bita River Ramsar Site as an Example of Science and Management Tools Working Together"

_sustainability, doi:10.3390/su13041775_

Round 1

Reviewer 1 Report

This work provides an interesting overview of environmental watershed management, from a socio-political point of view. The proposed scheme appears comprehensive and well thought out, but I feel that a number of key details are missing. In particular, more clarification on the calculation of simulated scenario results and definition of key terms are needed. These, and other points, are outlined below:

Lines 48-54: It would be helpful to have some kind of pictorial or graphical reference with this paragraph, perhaps a map showing dominant ecotypes int he basin or a plot showing precipitation and/or discharge trends.

Line 66: The criteria noted in this line needs to be stated or defined for clarity (i.e., what are the criteria and by whom were they defined?)

Line 74: I don't fully understand the wording "enabling conditions". Maybe it would be clearer to say "The work to establish environmental sustainability..."

Line 101: Can the "Orinoco Basin Report Card" be briefly described or at least, referenced?

Line 113: The term "Ramsar Site" should be defined and if possible, briefly described in more detail.

Line 122: The mentioned "stricter conservation methods" should be outlined or stated in more detail.

Line 131: What software, equations, and assumed variables were used in the modeling process? I think this is vital information.

Figure 3: The methodology, software, and/or equations, etc., behind this graph needs to be stated and detailed (or at least referenced, if based on previous work) in the methods.

Line 262: Can the rapidity of this deterioration be quantified somehow?

Line 263: This mention of dolphin population decline should be referenced.

Line 267: The differential gap analysis should be described in the methods in more detail.

Line 270: What specifically are the Ramsar Convention Criteria? This should be at least briefly described.

Line 271: Can more details on this participatory process be provided?

Lines 302-305: This sentence is a bit long, and difficult for me to understand due to its length. It may need to be subjected to more rigorous English editing.

Lines 311-313: Methods and software? I'm assuming GIS, but what about data sources?

Line 328: Pardon my ignorance, but I'm not so familiar with congrio and saladillo. I presume that these are tree types, but would it be possible for this to be stated for less familiar readers?

Author Response

Many thanks reviewer 1. Your comments and suggestions show us the key points to improve it. Here our answers to each point:

Lines 48-54: It would be helpful to have some kind of pictorial or graphical reference with this paragraph, perhaps a map showing dominant ecotypes int he basin or a plot showing precipitation and/or discharge trends.

A profile of the Bita river was included. We think that it provides a better panorama about the main ecosystems and their relation with flood dynamics.

Line 66: The criteria noted in this line needs to be stated or defined for clarity (i.e., what are the criteria and by whom were they defined?)

We have explained the Ramsar criteria here and into the section 3.4.

Line 74: I don't fully understand the wording "enabling conditions". Maybe it would be clearer to say "The work to establish environmental sustainability..."

We are agreed with this suggestion.

Line 101: Can the "Orinoco Basin Report Card" be briefly described or at least, referenced?

The ecosystem report card is a transformative assessment and communication product that compare environmental data to scientific or management thresholds as a vehicle for stakeholder engagement and shared visioning. We have included a short description and cited.

Line 113: The term "Ramsar Site" should be defined and if possible, briefly described in more detail.

All the paragraph was complemented as follow:

Ramsar site: assumes the implementation of a Ramsar site, that is an international designation under the Ramsar convention where the contracting parties (countries, in this case the Colombian government) are expected to manage and to maintain their ecological character and retain their essential functions and values for future generations and hence focuses on reducing the exploitation of natural resources and on curbing the historical trend of land cover change. Nevertheless, the establishment of the Ramsar site does not imply full conservation[33]. In fact, economic activities are allowed, but only those that do not cause harm to the environment. In this scenario we are assuming a proper management of natural resources without harm to it.

Line 122: The mentioned "stricter conservation methods" should be outlined or stated in more detail.

These actions include land use restrictions, prohibition of impact activities as fauna and flora hunting, mining and oil extraction, infrastructure construction and land ownership limitation. It was complemented into the text.

Line 131: What software, equations, and assumed variables were used in the modeling process? I think this is vital information.

References and descriptions were included in a new text. Equations and data used were referenced from a previous work.

Figure 3: The methodology, software, and/or equations, etc., behind this graph needs to be stated and detailed (or at least referenced, if based on previous work) in the methods.

Idem. But improving the references

Line 262: Can the rapidity of this deterioration be quantified somehow?

Yes, our analysis corroborate information reported by Mosquera-Guerra, where were estimated decrease of 35.7% in the number of dolphin individuals in six years for the Colombian Orinoco.

Line 263: This mention of dolphin population decline should be referenced.

Yes, our analysis corroborate information reported by Mosquera-Guerra, where were estimated decrease of 35.7% in the number of dolphin individuals in six years for the Colombian Orinoco.

Line 267: The differential gap analysis should be described in the methods in more detail.

It was complemented briefly.

Line 270: What specifically are the Ramsar Convention Criteria? This should be at least briefly described.

Same comment line 66. It was complemented.

Line 271: Can more details on this participatory process be provided?

All the stakeholders were mentioned in a new text.

Lines 302-305: This sentence is a bit long, and difficult for me to understand due to its length. It may need to be subjected to more rigorous English editing.

It was edited.

Lines 311-313: Methods and software? I'm assuming GIS, but what about data sources?

According with editor comments this senteces was changed

Line 328: Pardon my ignorance, but I'm not so familiar with congrio and saladillo. I presume that these are tree types, but would it be possible for this to be stated for less familiar readers?

  1. Yes they are trees congrio (Ascomium nitens) and saladillo (Cairapa llanorum). Line 325 and 326 are listed.

Reviewer 2 Report

This is an intreresting and well written report. However, it needs more refernces to support the observations after line 72. From this line onwards it makes a number declarations which I dont doubt are correct but as an academic publications there should be more reference to the sources of the data. 

Author Response

Many thanks for your comments. After a detail revision of all the reviewers and editor suggestion. We have now a new text that we hope fill the journal expectations. Additional references were cited to improve our conclusions comparing with academic researches.

Reviewer 3 Report

the authors simply describe the situation of the basin and the river and the importance of their protection. They allude to some models, but they absolutely do not explain what they are, how they work, how they were calculated or what data is included.

Specific comments:

Lines 30 and 70: Please explain what a Ramsar site is.

Line 40: The number 3 in m3/sec should be superscript and “sec” should be substituted with “s”.

Line 55: The full stop is missing.

Lines 69-72: Please check English in these sentences.

Line 79: After a semicolon, a full sentence should be reported (the verb is missing here).

Figure 1: Please check the font type and size of the years (2019-2020 is smaller than the others).

Lines 84-86: Please check English in this sentence.

Line 89: Please explain what a System Thinking approach is.

Line 99: Please explain what “seeing systems” are.

Line 102: Which indicators were taken into account?

Lines 118-121: Please check English in this sentence.

Lines 155-160: This part is not a result and should be first explained in the “Methods” section. Moreover, appropriate references should be added.

Lines 167-168: Is there a bullet point missing?

Lines 169-172: The use of (1) and (2) in this sentence is not clear.

Figure 3: What is a biodiversity index? How was it calculated for the different scenarios? What is “Dm nl” reported on the y-axis? It is quite strange that the index in previous years was constantly 1. Has biodiversity never changed in the past? Moreover, we are now at the end of 2020. It is nonsense to present the results of models starting from 2017. What are the updated conditions and the forecasts?

Lines 280-282: Please check English in this sentence.

Line 302: What are the two zoning schemes? In Figure 4 I can only see one zoning scheme, comprising three different zones (and it is not explained according to which parameters the zoning was made)

Author Response

The authors are thankful for your comments and suggestions. They have shown many points to improve and others coincide with others reviewers. It shows us your detail reading. Thanks.

Here a quick answers to your comments:

Lines 30 and 70: Please explain what a Ramsar site is.

It was included into the new text.

Line 40: The number 3 in m3/sec should be superscript and “sec” should be substituted with “s”.

We accept the suggestion conversion

Line 55: The full stop is missing.

we didn’t understand this comment, the sentece don't change

Lines 69-72: Please check English in these sentences.

English was reviewed

Line 79: After a semicolon, a full sentence should be reported (the verb is missing here).

The text was complemented as follow: at the same time, civil society organizations conformed the alliance such as

Figure 1: Please check the font type and size of the years (2019-2020 is smaller than the others).

The font was standardized

Lines 84-86: Please check English in this sentence.

Sentence improved

Line 89: Please explain what a System Thinking approach is.

This part of the manuscript was improved and now it has a new text.

Line 99: Please explain what “seeing systems” are.

Idem

Line 102: Which indicators were taken into account?

Indicators table was included.

Lines 118-121: Please check English in this sentence.

Done

Lines 155-160: This part is not a result and should be first explained in the “Methods” section. Moreover, appropriate references should be added.

This section will be improved according with comments from others reviewers. New cites were included

Lines 167-168: Is there a bullet point missing?

Yes, it has been corrected

Lines 169-172: The use of (1) and (2) in this sentence is not clear.

Yes, the numbers into the initial text tried to enumerate the main impacts of the reinforcing loops. But now, they are not suitable. In the new text they will be eliminated.

Figure 3: What is a biodiversity index? How was it calculated for the different scenarios? What is “Dm nl” reported on the y-axis? It is quite strange that the index in previous years was constantly 1. Has biodiversity never changed in the past? Moreover, we are now at the end of 2020. It is nonsense to present the results of models starting from 2017. What are the updated conditions and the forecasts?

A new cite was included. This part of the methodology was develop in a previous work. Include all the details will change the character and scope of this paper. For this reason we improved the methodological framework and cited the details.

Lines 280-282: Please check English in this sentence.

Edited

Line 302: What are the two zoning schemes? In Figure 4 I can only see one zoning scheme, comprising three different zones (and it is not explained according to which parameters the zoning was made)

Edited.

Round 2

Reviewer 1 Report

This article is much improved with greater clarity, and I think it is fit for publication in its present form. I would still recommend a last round of English grammar editing, as some minor errors still exist, but nothing severe.

Author Response

Thank you for your comments. The english version is under a deep revision.

Thanks again.

Cesar

Reviewer 2 Report

Much better explanation and report

Author Response

We appreciate your suggestions to improve the manuscript. Thanks for your feedback.

Some english editions are under revision.

Authors

Reviewer 3 Report

I am still a bit skeptical about the whole paper idea, but I recognise that the authors made an effort to properly reply to the reviewers' comments and to improve the manuscript.

Author Response

First of all, We would like to thank for your proper suggestions and comments to improve the manuscript. As you mention, we have tried to follow different advices from different points of view, especially some gaps that were basics. And we consider that now it is much better. In some point I am agree with you. However, is important to highlight that it is not a research work per se. The manuscript compiles different technical works and reseaches that have triggered to increase freshwater and biodiversity protection. Some of them only were mentioned and in some point (as I mention) is not easy chain them.

A final english edition is under review.

Thanks again for your comments.

Authors